# Dysregulated plasma lipid mediator profiles in critically ill COVID-19 patients

Francesco Palmas[1], Jennifer Clarke[2], Romain A. Colas[1], Esteban A. Gomez[1], Aoife Keogh[2], Maria Boylan[2], Natalie McEvoy[2], Oliver J. McElvaney[2], Oisin McElvaney[2], Razi Alalqam[2], Noel G. McElvaney[2], Gerard F. Curley[2‡], Jesmond Dalli[1,3‡]*

1 William Harvey Research Institute, Barts and The London School of Medicine and Dentistry, Queen Mary University of London, London, United Kingdom, 2 Department of Anaesthesia and Critical Care, Royal College of Surgeons, Dublin, Ireland, 3 Centre for Inflammation and Therapeutic Innovation, Queen Mary University of London, London, United Kingdom

☉ These authors contributed equally to this work.
‡ These authors are joint senior authors on this work.
* j.dalli@qmul.ac.uk

**Data Availability Statement:** All relevant data are within the manuscript and its Supporting Information files.

## Abstract

Coronavirus disease (COVID)-19, as a result of Severe Acute Respiratory Syndrome Coronavirus 2 (SARS-CoV-2) infection, has been the direct cause of over 2.2 million deaths worldwide. A timely coordinated host-immune response represents the leading driver for restraining SARS-CoV-2 infection. Indeed, several studies have described dysregulated immunity as the crucial determinant for critical illness and the failure of viral control. Improved understanding and management of COVID-19 could greatly reduce the mortality and morbidity caused by SARS-CoV-2. One aspect of the immune response that has to date been understudied is whether lipid mediator production is dysregulated in critically ill patients. In the present study, plasma from COVID-19 patients with either severe disease and those that were critically ill was collected and lipid mediator profiles were determined using liquid chromatography tandem mass spectrometry. Results from these studies indicated that plasma concentrations of both pro-inflammatory and pro-resolving lipid mediator were reduced in critically ill patients when compared with those with severe disease. Furthermore, plasma concentrations of a select group of mediators that included the specialized pro-resolving mediators (SPM) Resolvin (Rv) D1 and RvE4 were diagnostic of disease severity. Interestingly, peripheral blood SPM concentrations were also linked with outcome in critically ill patients, where we observed reduced overall concentrations of these mediators in those patients that did not survive. Together the present findings establish a link between plasma lipid mediators and disease severity in patients with COVID-19 and indicate that plasma SPM concentrations may be linked with survival in these patients.

## Introduction

Acute Respiratory Distress Syndrome (ARDS) is a life-threatening consequence of Severe Acute Respiratory Syndrome Coronavirus 2 (SARS-CoV-2) infection. The syndrome is

**Funding:** This work was supported by funding from the European Research Council (ERC) under the European Union's Horizon 2020 research and innovation programme (grant no: 677542) and the Barts Charity (grant no: MGU0343) to JD. JD is also supported by a Sir Henry Dale Fellowship jointly funded by the Wellcome Trust; the Royal Society (grant 107613/Z/15/Z). The funders had no role in study design, data collection and analysis, decision to publish, or preparation of the manuscript.

**Competing interests:** The authors have read the journal's policy and the authors of this manuscript have the following competing interests: JD is scientific founder and director of Resolomics Ltd. This does not alter our adherence to PLOS ONE policies on sharing data and materials. There are no patents, products in development or marketed products associated with this research to declare.

characterised by hypoxemia, diffuse bilateral pulmonary infiltrates, and reduced respiratory system compliance [1, 2]. The mainstay of treatment is mechanical ventilation using specific strategies to avoid ventilator-induced lung injury and lung strain [3]. Prior to the emergence of SARS-CoV-2, the incidence of ARDS amongst patients in Intensive Care Units (ICUs) was estimated to be 10%, with a mortality rate of 40% in those with severe disease [4].

Reports suggest that among those infected with SARS-CoV-2, up to 20% of patients develop severe disease requiring hospitalization, while approximately 5%-8% of the total infected population needs ICU admission [5, 6]. The CDC outlines over 440,000 deaths secondary to SARS-CoV-2 in the United States (https://covid.cdc.gov/covid-data-tracker/#cases_casesper100klast7days), while UK mortality rates currently are reported at 112,660 (https://coronavirus.data.gov.uk), making this now a leading worldwide cause of morbidity and mortality. As well as the impact on patients' lives, increased demand for critical care beds and mechanical ventilators has a significant socio-economic impact and places increased pressure on an already over-burdened healthcare system.

Although, the RECOVERY trial demonstrated dexamethasone to be efficacious in reducing mortality in patients with severe disease [7] and Remdesivir has been shown to reduce length of hospital stay [8], the treatments for SARS-CoV-2 remain limited and the mechanisms that underlie these strategies are incompletely understood. It seems that development of severe disease is not solely related to viral load and could involve a delayed and excessive inflammatory response [9]. Several groups have aimed at characterising this aspect in SARS-CoV-2 infection and delineate distinctions in the inflammatory profile between SARS-CoV-2-related ARDS compared with non-SARS-CoV-2 ARDS or other causes of respiratory failure requiring ICU admission [10, 11]. However, many key questions remain unanswered, such as the degrees to which viral load and host response affect disease severity and the immunologic mechanisms behind this condition.

One aspect of the immune response that has, to date, been greatly understudied in COVID-19 patients is the role of lipid mediators (LM). LM are autacoids produced from essential fatty acids and constitute a central part of the concerted immune response. They are involved in all aspects of the inflammatory response; from initiation, propagation, and resolution of inflammation [12, 13]. While the resolution of inflammation was previously thought to be a passive process, it has since been established that specific LM are central players in driving endogenous counter-regulation of inflammation and activation of resolution [14, 15].

Arachidonic acid (AA), a poly-unsaturated fatty acid (PUFA), is the precursor for the pro-inflammatory prostaglandins (PG), leukotrienes (LT), and thromboxanes (Tx) as well as the pro-resolving lipoxins (LX). The omega-3 essential fatty acids eicosapentaenoic (EPA), docosapentaenoic (n-3 DPA), and docosahexaenoic (DHA) are precursors to three families of specialized pro-resolving mediators (SPM), namely protectins, resolvins, and maresins. These molecules play a pivotal role in regulating viral replication as well as in reprograming the host innate and adaptive immune response [16–18]. Through the activation of cellular receptors, each SPM exerts both cell and organ-specific properties that allow them to limit the infiltration of polymorphonuclear neutrophils and promote efferocytosis [15, 19].

The role of SPMs in bacterial and viral infection has been well studied [19, 20] in animal models of other viral infection including influenza and herpes simplex viruses, where SPMs have been shown to reduce disease severity [21, 22]. Of note, one study explored the role of LM in coronavirus infection, namely SARS-CoV. Here, the authors demonstrated that an increase in pro-inflammatory eicosanoids following increased PLA2G2D levels in the lungs, led to worse outcomes in SARS-CoV infected mice [23]. Other studies have identified the utility of SPMs in promoting alveolar fluid clearance and the resolution of acute lung injury [24, 25].

For these reasons, in the present study we aimed at characterising the LM profiles of patients with COVID-19 in order to determine the potential relationships between plasma LM levels and disease severity, as well as outcome in these patients.

## Materials and methods

### Study setting and design

Between 16th March 2020 and 1st May 2020, 38 adult (age > 18 years) patients with confirmed SARS-CoV-2 infection by viral PCR were recruited from Beaumont Hospital for an observational cohort study with peripheral blood lipid mediator profiling and analysis of clinical outcomes. Blood sampling was performed within 24 h of recruitment.

Patients above 18 years of age were approached for informed consent to blood sampling by the research team if they met the criteria of a positive PCR result for SARS-CoV-2. For patients in the ICU lacking capacity, this was sought from their next of kin under appropriate ethical approval. Informed consent was obtained retrospectively from these patients, where possible.

The cohort included patients admitted to the wards and ICU with SARS-CoV-2 infection. The majority of patients were recruited within 72 h of a positive PCR result. Due to the prospective nature of our sampling, we were able to capture a heterogenous population of ward patients and ICU patients, some recruited before admission and during admission to ICU. Patients were catagorised into two main groups, critically ill and severe disease. Patients were categorised as critically ill if they required invasive mechanical ventilation in the ICU. The severe group was defined by those who were managed at ward level with supplemental oxygen or non-invasive ventilation. Plasma was obtained from peripheral blood by means of centrifugation at 1500 g for 10 min at room temperature and then frozen before shipment. Demographics, age, gender, BMI, $PaO_2/FiO_2$ ratio, Sequential Organ Failure Assessment Score, d-dimers, ferritin, ICU length of stay and survival were recorded for the cohort.

Ethical approval was received from the Beaumont Hospital Ethics Committee (REC #18/52, 17/06).

### Lipid mediator profiling

Samples were thawed and 4 mL of ice-cold methanol containing deuterium-labelled internal standards, representing each region of the chromatographic analysis (500 pg for $d_4$-PGE$_2$, $d_8$-5-HETE, $d_4$-LTB$_4$, $d_5$-LXA$_4$, and $d_5$-RvD2, 250 pg for $d_5$-MaR1, $d_5$-MaR2, and $d_5$-RvD3, 100 pg for $d_5$-RvE1, and 25 pg for $d_5$-17R-RvD1), were added to facilitate analytes identification and quantification. Samples were then stored at -20˚C for a minimum of 45 min to allow for protein precipitation. Subsequently, supernatants were subjected to solid phase extraction, collecting methyl formate and methanol fractions. Following solvent evaporation, samples were resuspended in methanol/water (1:1, vol/vol) phase for injection on a Shimadzu LC-20AD HPLC and a Shimadzu SIL-20AC autoinjector, coupled with QTrap 6500+ or QTrap 5500 (ABSciex). For the analysis of unconjugated lipid mediator eluted in methyl formate fraction, Agilent Poroshell 120 EC-C18 column (100 mm x 4.6 mm x 2.7 μm) was kept at 50˚C and mediators eluted using a mobile phase consisting of methanol (0.01% acetic acid)/water (0.01% acetic acid) of 20:80 (vol/vol) that was ramped to 50:50 (vol/vol) over 0.5 min and then to 80:20 (vol/vol) from 2 min to 11 min, maintained till 14.5 min and then rapidly increased to 98:2 (vol/vol) for the next 0.1 min. This was subsequently maintained at 98:2 (vol/vol) for 5.4 min. Flow rate was maintained at 0.50 mL/min. In the analysis of peptide-lipid conjugated mediators eluted in the methanol fraction, Agilent Poroshell 120 EC-C18 column (100 mm × 4.6 mm × 2.7 μm) was kept at 50˚C and mediators eluted using a mobile phase consisting of methanol (0.5% acetic acid)/water (0.5% acetic acid) at 55:45 (vol/vol) over 5 min, that

was ramped to 80:20 (vol/vol) for 2 min, maintained at 80:20 (vol/vol) for the successive 3 min and ramped to 98:2 (vol/vol) over 3 min. This condition was kept for 3 min. A flow rate of 0.6 mL/min was used throughout the experiment. Both QTrap 6500+ and QTrap 5500 were operated using a multiple reaction monitoring (MRM) method and positive and negative acquisition mode, respectively, as previously reported [26]. Each lipid mediator was identified using established criteria, these included: 1) matching retention time to synthetic or authentic standards with maximum drift between the expected retention time and the observed retention time of ± 0.05 min, 2) presence of a peak with a minimum area of 2000 counts, 3) at least 4 data points, and 4) matching of at least 6 diagnostic ions to that of reference standard, with a minimum of one backbone fragment being identified [26]. Calibration curves were obtained for each mediator using lipid mediator mixtures at 0.78, 1.56, 3.12, 6.25, 12.5, 25, 50, 100, and 200 pg that gave linear calibration curves with an $r^2$ values of 0.98–0.99.

## Statistical analysis

Univariate statistical analyses were performed using R (https://www.r-project.org/), Prism 8, and Microsoft Excel, while multivariate analyses were conducted using R and MetaboAnalyst 4.0 [27]. Differences between two groups were determined using two-tailed Mann-Whitney test for non-normal data distribution and significance was considered as P-value < 0.05. Differences among more than two groups were calculated using Kruskal-Wallis statistical for non-normal data distribution and significance was determined using Benjamini Hochberg correction and an adjusted P-value < 0.05.

Partial Least Squares-Discriminant Analysis (PLS-DA) was performed using MetaboAnalyst 4.0 and applying autoscaling on lipid mediator concentrations, together with Leave One Out Cross Validation (LOOCV) method. PLS-DA builds a multivariate model that identifies a direction to explain variance among variables (lipid mediator concentrations) and classify the observations (samples). During classification, each sample is assigned with a score for each of the two coordinates, representative of a combination of independent variables (principal component, PC), and it is plotted as a single point in a two-dimensional scores plot. This approach helps to visualise the clustering of samples based on profile similarity. Variance Importance of Projection (VIP) is a measure of the relative contribution of the different variables, highlighting which are the most relevant to the separation of observations into classes (*e.g.* pathological status).

Least Absolute Shrinkage and Selection Operator (LASSO) regression model was performed using the R package "glmnet" (https://cran.r-project.org/web/packages/glmnet/index.html). This machine learning method performs both variable selection and regularization in order to improve predictivity and interpretability of the statistical models it produces [28]. Here, the variables contributing the least to the prediction model are forced to a coefficient value equal to zero, while the most relevant features are conserved. To create this model, we first defined the value of λ (*i.e.* the numeric value defining how simple the model could be) that minimises the cross-validation prediction error without losing relevant features, known as λmin. With λmin, only a subset of variables is used successfully in the predictive model by fixing the sum of all regression coefficients lower than a threshold. This results in a compromise between prediction and simplicity of the model.

## Results

### Patient characteristics

During the study period, plasma samples were obtained from 23 critically ill patients with SARS-CoV-2 infection and from 15 patients with severe disease. Baseline characteristics are

**Table 1. Demographics and clinical characteristics of COVID-19 patients with severe disease and critically ill patients.**

| | Critically ill (n = 23) | Severe (n = 15) |
|---|---|---|
| Age (years) | 51 (IQR 42–60) | 70 (IQR 44–73) |
| Male, No. (%) | 17 (74) | 8 (53) |
| D-Dimer (ug/ml) | 0.79 (IQR 0.52–1.26) | 1.6 (IQR 0.6–2.4) |
| Ferritin (ng/ml) | 1260 (IQR 788–1260) | 1434 (IQR 569.5–1948) |
| PaO2/FiO2 Ratio(mmHg) | 142.5 (IQR 120–165) | 237.8 (IQR 206.6–253.9) |
| WHO Ordinal Scale for Clinical Improvement Score | 7 (IQR 7–7) | 5 (IQR 4–8) |
| Death, No. (%) | 4 (18.2)* | 4 (26.7) |

Continuous variables are represented as median (interquartile range) and categorical variables are represented as absolute numbers (percentages).

* *Data available for n = 22.*

outlined in Table 1. Of note, critically ill patients were younger than those with severe disease, median age 51 (IQR 42–60) in critically ill group and 70 (IQR 44–73) in severe. Those in the critically ill group also had lower baseline $PaO_2/FiO_2$ ratios when compared to their severe counterparts, 142.5 (IQR 120–165) versus 237.8 (IQR 206.6–253.9). Those in the critically ill group were also noted to have higher scores on the WHO Ordinal Scale for Clinical Improvement [29] compared to those in the severe group, 7 (IQR 7–7) versus 5 (IQR 4–8) respectively.

Critically ill patients whose outcomes were known (n = 22) were then further categorised into survivors and non-survivors (Table 2). The median age amongst those who survived to discharge was 49 (IQR 44–60) compared to 55 (IQR 38–70) in non-survivors. Sequential organ failure assessment scores on admission to ICU were the same between groups, median score 8 (IQR 6–9) in survivors and 8 (IQR 5–11) in non-survivors. WHO Ordinal Scale for Clinical Improvement Scores were higher in non-survivors as compared to survivors, 8 (IQR 8–8) versus 7 (IQR 7–7). Median $PaO_2/FiO_2$ ratios on admission to ICU were also similar between groups 150 (IQR 125.6–178.1) in survivors versus 146.3 (IQR 123.8–163.1) in non-survivors.

## Upregulation of LM in patients with moderate disease

Using LC-MS/MS, we profiled LM in plasma from these patients and identified LM from all four bioactive metabolomes, including n-3 DPA and DHA metabolomes, in accordance with

**Table 2. Demographics and clinical characteristics of survivors and non-survivors within COVID-19 critically ill patients.**

| | Survivors (n = 18) | Non-Survivors (n = 4) |
|---|---|---|
| Age (years) | 49 (IQR 44–60) | 55 (IQR 38–70) |
| Male No. (%) | 12 (66) | 4 (100) |
| BMI (kg/m$^2$) | 34.6 (IQR 29.7–41.6) | 30.1 (IQR 23.1–39.6) |
| D-Dimer (ug/ml) | 0.74 (IQR 0.45–1.20) | 4.06 (IQR 0.67–37.52) |
| Ferritin (ng/ml) | 1127 (IQR 730–2105) | 1286 (IQR 1005–4509) |
| ICU Length of Stay (days) | 9 (IQR 7.75–17.25) | 12.5 (IQR 3–20) |
| SOFA Score | 8 (IQR 6–9) | 8 (IQR 5–11) |
| PaO2/FiO2 Ratio (mmHg) | 150 (IQR 125.6–178.1) | 146.3 (IQR 123.8–163.1) |

Continuous variables are represented as median (interquartile range) and categorical variables are represented as absolute numbers (percentages). Abbreviations: SOFA: Sequential Organ Failure Assessment.

published criteria [26]. PLS-DA, a multivariable linear regression model that uses variables (*i. e.* LM) to separate experimental groups of observations (*i.e.* samples), was used to highlight differences in LM concentrations between the two groups. Different LM clusters were observed between plasma samples from patients with severe disease compared with critically ill patients as depicted in the score plot shown in Fig 1A. Evaluation of the VIP scores, which identify the LM that most strongly contribute to the observed separation, identified 18 LM with a VIP score >1 (Fig 1B). Notably, the majority of LM was found to be differentially expressed between the two groups and were upregulated in plasma of patients with severe disease. Among the mediators upregulated in plasma from these patients were members of the pro-inflammatory LT family, including $LTB_4$, $LTD_4$, $LTE_4$, and $LTC_4$, and the further metabolite of the potent pro-thrombogenic and immunosuppressive mediator $TxA_2$, $TxB_2$. Additionally, we also found increased concentrations of SPM in plasma from patients with severe disease that included RvD1, MCTR1, RvT3, and $PD1_{n-3\ DPA}$ (S1 Table).

LM pathway analysis was then performed to better evaluate potential differences in LM biosynthetic pathways between the two patient groups (Fig 2). This analysis highlighted that the majority of LM downregulated in critically ill patients (including LT, RvD1, and RvD3) were derived from 5-lipoxygenase (ALOX5), thereby suggesting a downregulation in the activity of this enzyme with increasing disease severity.

Having observed a shift in LM profiles between the two patient groups, we next investigated whether any LM could predict disease severity by using LASSO regression analysis [28]. This analysis identified a subset of 7 LM, namely $TxB_2$, $LTD_4$, $RvE_4$, 20-COOH-$LTB_4$, 20-OH-MaR1, RvD1, and RvD3, that provided an 87% accuracy in identifying disease severity in these patients (Fig 3). Together, these findings demonstrate that LM levels become dysregulated with decreasing disease severity, a shift linked with a downregulation in ALOX5 activity.

## Higher SPM levels are linked with a better outcome in critically ill patients

Having observed differences in LM profiles between patients with severe disease and those that were critical, we next examined whether these profiles were also reflective of outcome (*i. e.* discharged *vs.* deceased) in critically ill patients. Among the 23 critical patients recruited into this study, outcome was known for 22 subjects, where 18 were discharged from hospital and 4 died (Table 2). Given the central role that SPM play in host protection and the regulation of inflammation, we evaluated the overall SPM concentrations in plasma collected within 24h from admittance to the ICU in the two patient groups. Fig 4 displays the mean SPM concentrations as well as the ratio between the means of SPM to pro-inflammatory eicosanoids (SPM/Pro) as a measure of the resolution status of these patients [30]. Both were significantly decreased in deceased patients in comparison to discharged patients. Fig 4D shows differences in LM profiles between the two patient groups as further highlighted using PLS-DA, where a separation was observed between the patients' clusters. Assessment of the corresponding VIP scores indicated differential regulation of both SPM and pro-inflammatory mediators between the groups, with an upregulation of pro-inflammatory eicosanoids such as $PGE_2$, and the further metabolite of $LTB_4$, *i.e.* 20-OH-$LTB_4$, in non-survivors (Fig 4D and S2 Table).

In order to evaluate whether specific LM biosynthetic pathways are differently regulated in non-survivors when compared with survivors, we performed a pathway analysis focusing on those mediators found to be differentially modulated between the two groups (VIP scores >1). Amongst the obtained results, a striking observation was a marked increase in the production of AA-derived mediators from both the ALOX5- and COX pathways in non-survivors when

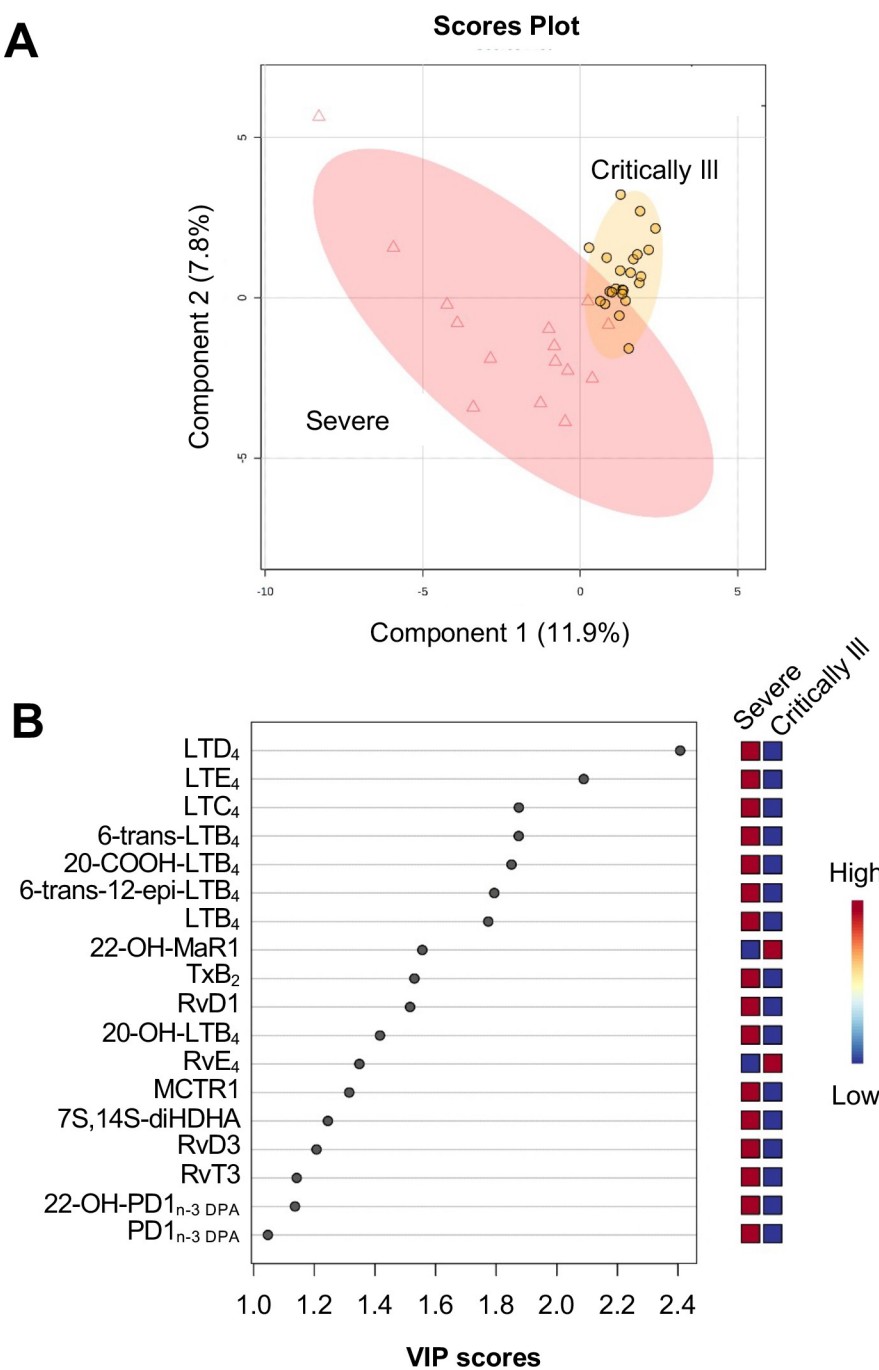

**Fig 1. Upregulation of plasma LM concentrations in patients with severe disease.** Plasma was collected from COVID-19 patients with severe disease (n = 15) and critically ill patients (n = 23) with 24h of admittance. LM identified and quantified using lipid mediator profiling and concentrations were evaluated using PLS-DA, generating (A) scores plot displaying separation between the two groups and (B) VIP scores of 18 LM with the greatest contribution to group separation.

compared with survivors (15R-LXA$_4$, 6-trans-LTB$_4$ and PGE$_2$. Fig 5). These findings suggest that alterations in LM biosynthesis, leading to an overall loss in SPM formation, contributes to a worse outcome in patients with severe disease.

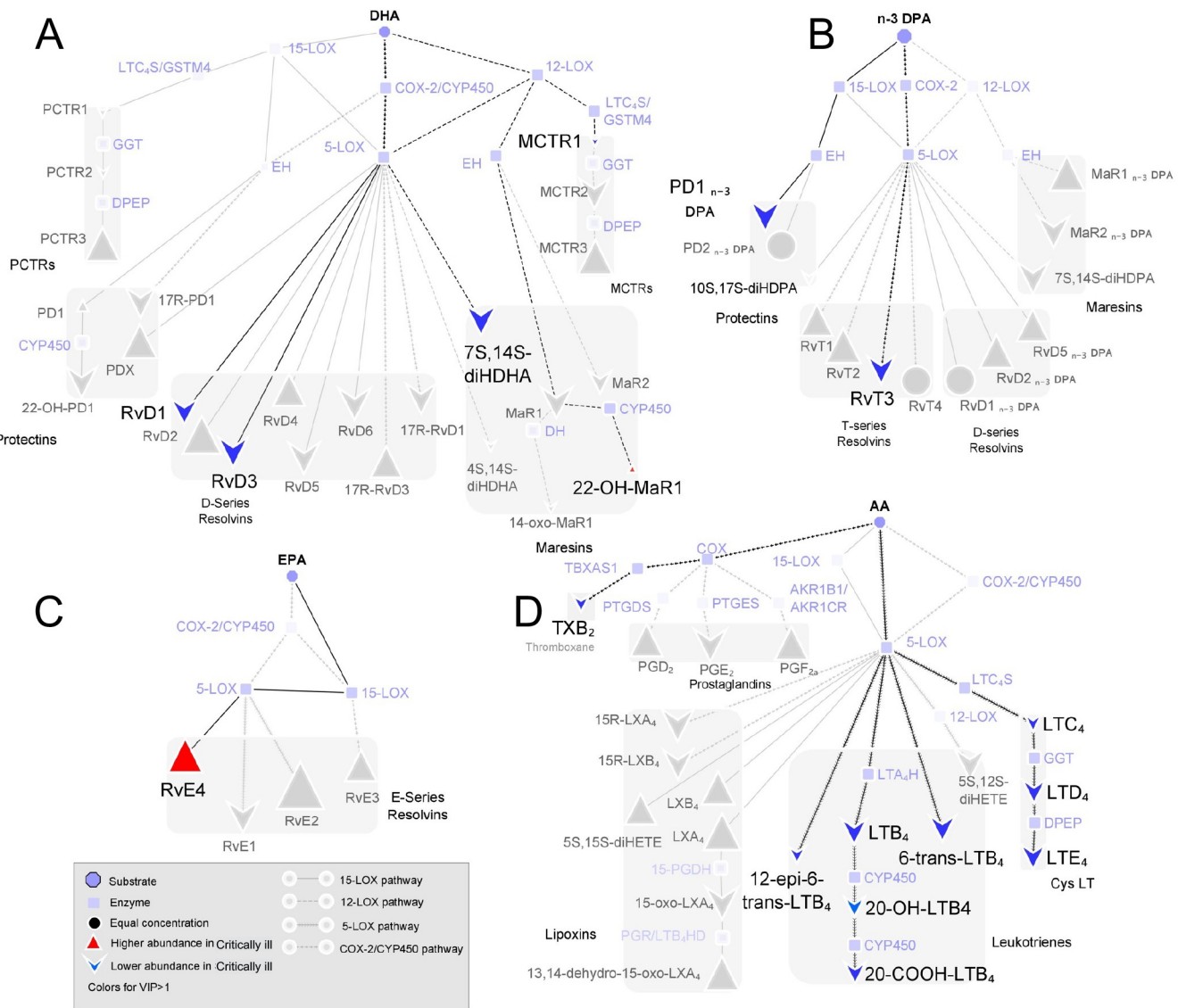

**Fig 2. LM pathway analysis identifies an upregulation in ALOX5 activity in patients with severe disease.** Pathway analysis highlighting those biosynthetic pathways liked with mediators found to contribute to the separation between patients with severe symptoms and those that were critically ill in the PLS-DA analysis (VIP Scores >1) for mediators from the (A) DHA, (B) n-3 DPA, (C) EPA, and (D) AA bioactive metabolomes. Results expressed as fold-change differences vs concentrations in plasma from patients with severe disease. Results are representative of n = 23 critically ill patients and n = 15 patients with severe disease.

## Discussion

In this study, using a systematic approach, we investigated the peripheral blood LM concentrations in patients diagnosed with SARS-CoV-2 infection. In critically ill patients, we observed a down-regulation of LM biosynthetic pathways, characterised by a decrease in both pro-inflammatory eicosanoids and SPM. In addition, comparison of LM concentrations between critically ill patients that were discharged *vs.* those that did not survive demonstrated that higher plasma SPM concentrations were linked with survival. Altogether, these findings suggest that disruptions in LM biosynthesis, and in particular SPM, are linked with disease severity and outcome in patients with SARS-Cov2 infections.

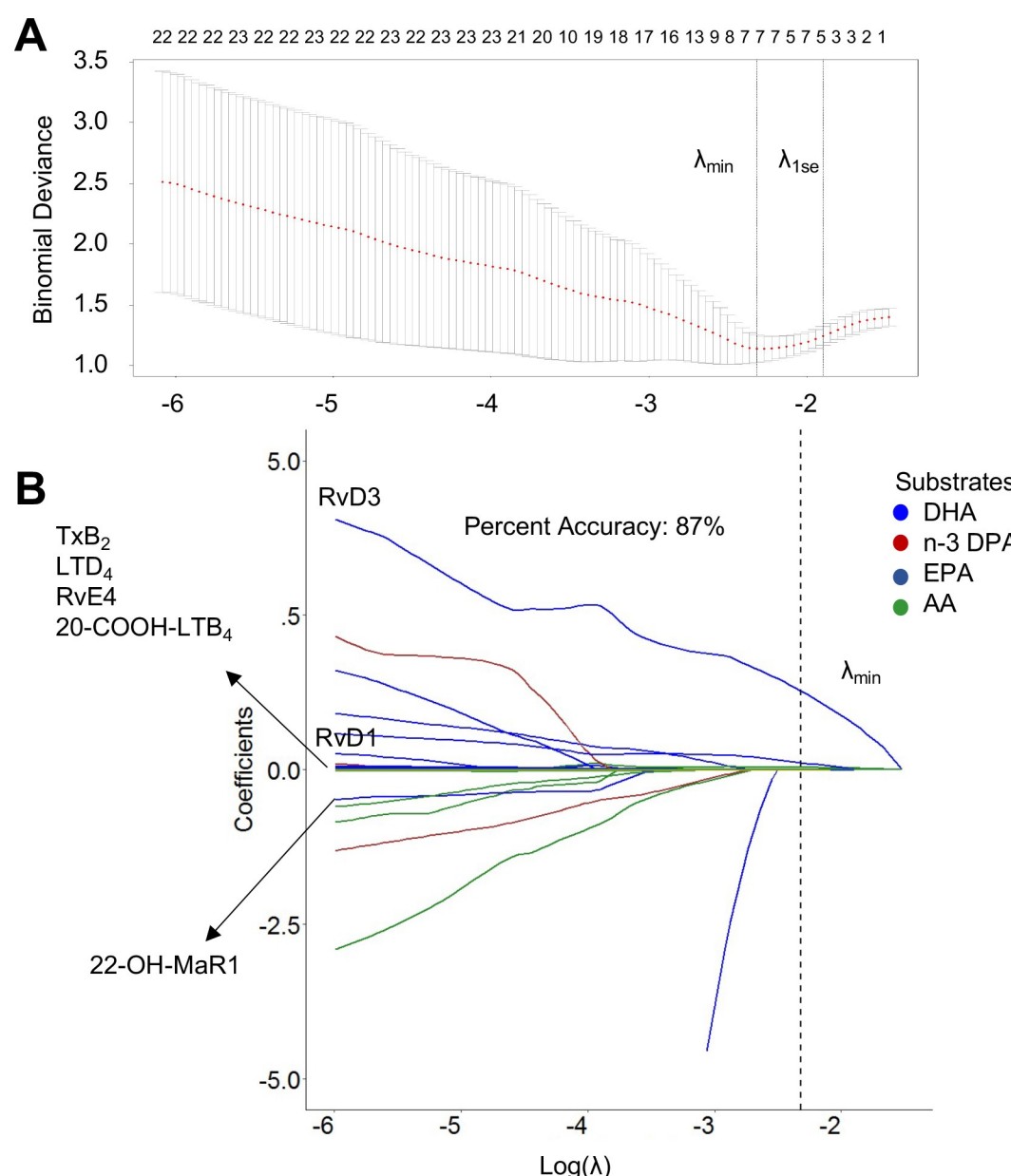

**Fig 3. Plasma LM concentrations are diagnostic of disease severity.** LASSO regression model (A) identified 7 LM (RvD3, RvD1, 22-OH-MaR1, TxB$_2$, LTD$_4$, RVE$_4$, and 20-COOH-LTB$_4$) as the minimum number of variables based on λxsmin to create the most stable model (B) with a percentage of accuracy of 87%. The model was validated by 10-fold cross validation. LM are represented with lines of different colours based on their belonging metabolomes; blue DHA, red n-3 DPA, dark blue EPA, and green AA.

SARS-Cov-2 infection leads to a wide range of illness severity, ranging from asymptomatic to ARDS requiring mechanical ventilation. For these reasons, significant effort has gone into understanding the cellular and molecular mechanisms that lead within such a spectrum. Notably, underlying health conditions and in particular chronic inflammatory diseases are associated with a worse prognosis [31]. While the mechanisms driving disease severity remain unclear, the clinical association of inflammatory mediators such as IL-6 and lactate

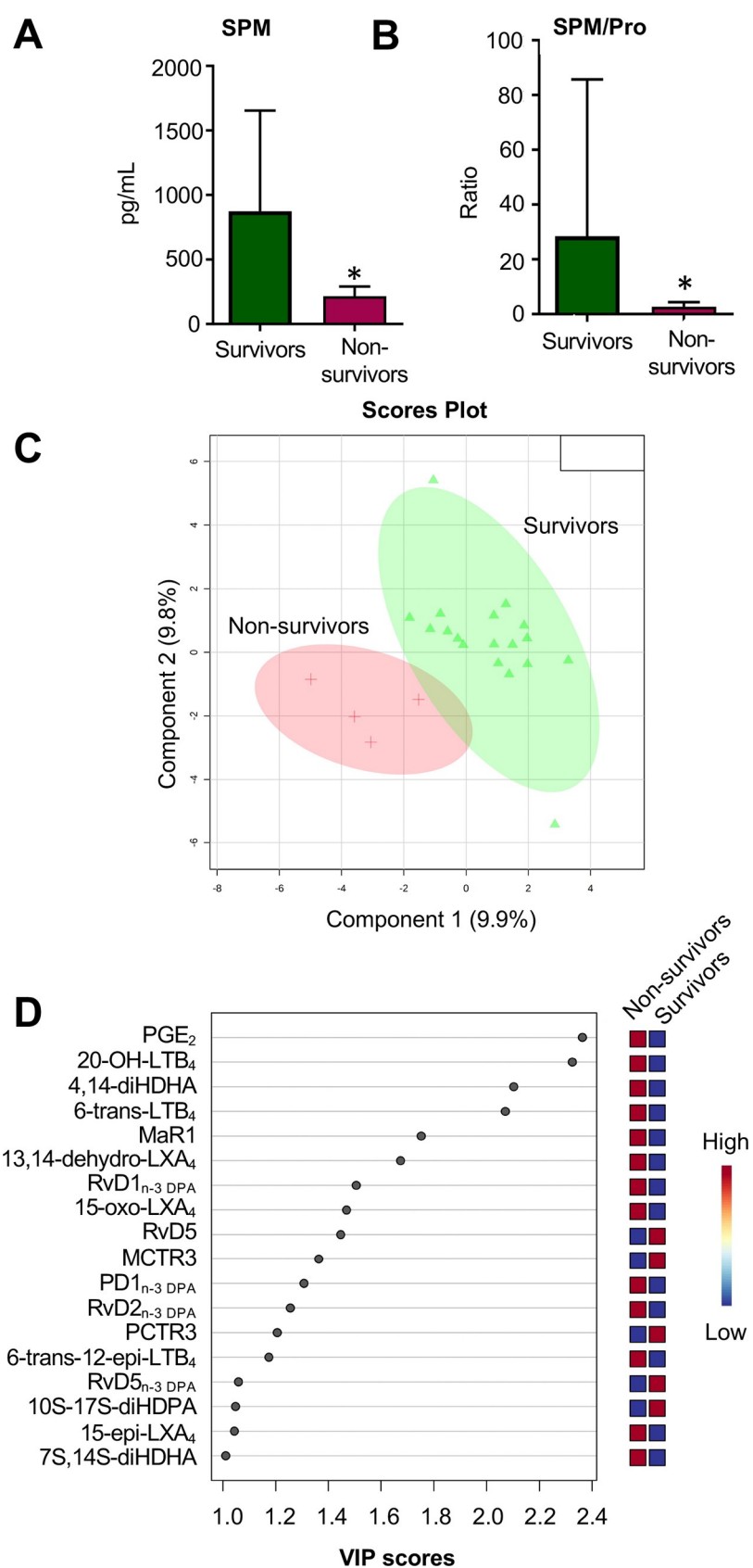

**Fig 4. Increased plasma SPM concentrations are linked with survival in critically ill patients.** Plasma was collected from critically ill patients within 24h of admittance to the ICU and plasma lipid mediators were identified and quantified using lipid mediator profiling. (A) Cumulative plasma SPM concentrations of (B) Ratio of SPM to pro-inflammatory eicosanoids (Pro–LT, PG, TXB$_2$). Results are mean ± SD. N = 18 for survivors and 4 for non-survivors. Statistical differences were evaluated using Two-tailed Mann-Whitney test for non-normal data distribution and significance was considered as P-value * < 0.05. (C,D) LM profiles were evaluated using PLS-DA generated (C) score plot displaying separation between the two groups and (D) VIP scores of 18 LM with the greatest contribution to group separation. Results are representative of n = 18 for survivors and 4 for non-survivors.

dehydrogenase (LDH) with severe cases suggests that excessive inflammation is reflective of a poor clinical outcome [32–34].

In the present study, we found that overall plasma LM concentrations were drastically reduced in critically ill patients when compared with those that had severe disease. Combining

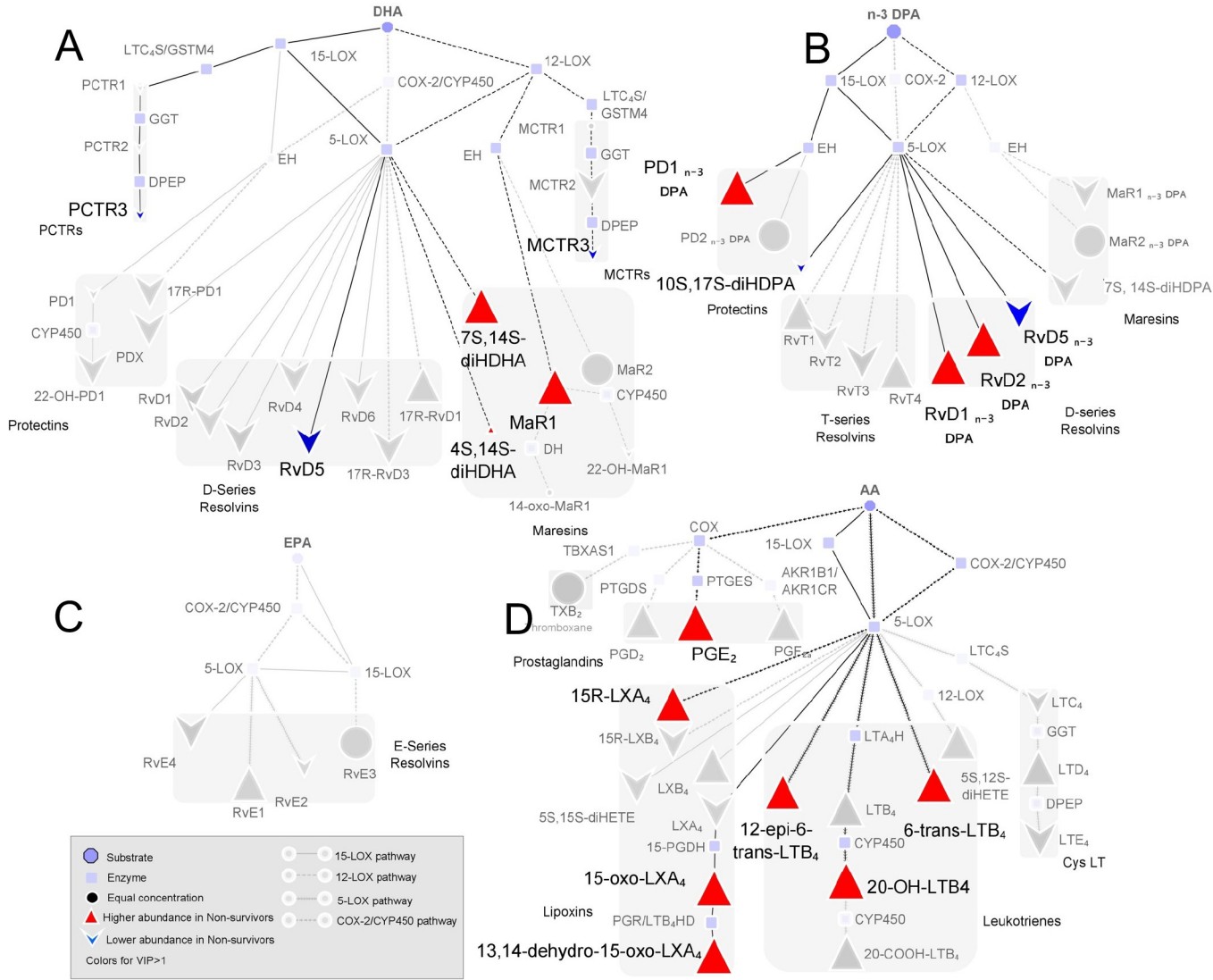

**Fig 5. LM pathway analysis highlights upregulation of AA-derived LM in non-survivors patients.** Pathway analysis highlighting those biosynthetic pathways liked with mediators found to contribute to the separation between survivors and non-survivors groups in the PLS-DA analysis (VIP Scores >1) for mediators from the (A) DHA, (B) n-3 DPA, (C) EPA, and (D) AA bioactive metabolomes.

these observations with the enhanced systemic inflammation of the non-surviving patients, it is suggested that dysregulated LM biosynthesis is implicated in disease progression. This relation is further supported by the markedly reduced SPM concentrations in non-surviving patients when compared with those eventually discharged. Similarly, in patients with meningeal tuberculosis LM concentrations in cerebrospinal fluids correlated with survival and lower LM levels were observed in non-surviving patients when compared with surviving ones [35].

Lipid mediators are involved in both the initiation and perpetuation of phlogistic events as well as in the termination of acute inflammation. Among the eicosanoids the PGs, LTs and CysLTs are important in activating the immune response following tissue injury; PGs and CysLTs promote vascular leak, facilitating cellular egress to the insult site, while LTB$_4$ acts as a potent chemoattractant. Excessive production of these molecules is associated with the propagation of both local and systemic inflammation through dysregulated vascular responses and leukocyte activation and recruitment. In addition, while CysLTs' display ionotropic actions and promote smooth muscle contraction in the lungs, PGs display immunosuppressive actions by hindering effector T-cell responses, a relevant mechanism in other SARS-CoV infections [23]. On the other hand, the SPM, display potent immune-regulatory actions during viral infections, whereby RvE1 regulates pathogenic T-cell and neutrophil influx during herpes simplex inflections and limits corneal neovascularization [22]. In influenza infections, the precursor to the DHA-derived resolvins, 17-HDHA, upregulates the production of neutralizing antibodies to the virus [36], whereas PDs family inhibits influenza virus replication *via* RNA export machinery [21]. SPM also regulate eicosanoid formation, a mechanism that is at least in part reliant on the ability of these molecules to switch the activity of enzymes involved in the formation of these molecules. For instance, LXA$_4$ limits LTB$_4$ formation in macrophages by preventing the phosphorylation of ALOX5, hence its translocation to the nuclear envelope. This mechanism is also implicated in the upregulation of SPM because the retention of ALOX5 in the cytosol leads to its coupling with ALOX15, in turn upregulating resolvins formation. Furthermore, SPM reduce the production of pro-inflammatory cytokines, including those involved in SARS-Cov-2 such as IL-6 and IL-1β [11, 37].

Herein we found a marked downregulation of both SPM and pro-inflammatory eicosanoids in critically ill patients compared with those with severe disease. Among SPM, we highlighted a decrease in RvD1 and RvD3 which, together with their epimeric forms carrying analogous biological functions, display potent lung protective actions in experimental models of lung inflammation and bacterial infections *via* the regulation of innate host immune responses, counter-regulation of inflammatory cytokine production, and regulation of epithelial cell biology [38–40]. Moreover, RvD1 limits tissue oxidative stress and NF-kB activation [13]. We also found a reduction in PD1$_{n-3\ DPA}$, a member of the n-3 DPA-derived PDs, which also displays potent immune-regulatory actions on monocytes and macrophages, both of which have been implicated in the propagation of systemic inflammation in COVID-19 patients [41]. Interestingly, in critically ill patients we found both a dysregulation of SPM formation. Assessment of the biosynthetic pathways found to be altered highlighted a disruption in the activity of ALOX5 in these patients. This observation is also in line with findings made in patients with meningeal tuberculosis [35]. Notably, Schwarz and colleagues reported an upregulation of ALOX5 expression in COVID-19 patients when compared with severe disease [42]. This suggests that upregulation in the activity/expression of this enzyme is a protective response that when dysregulated may lead to a poor outcome.

One of the challenges in treating patients with SARS-CoV-2 infections is the lack of robust biomarkers that will predict severity of disease and disease course. Early diagnosis of severe infection could aid earlier initiation of targeted life-saving treatments such as immune-modulation. In this study, we investigated the utility of LM to differentiate patients with severe life-

threatening disease from patients with milder disease in hospital, finding that indeed a subset of these mediators were strong predictors of disease severity (accuracy 87%). Given that LM are of great importance in orchestrating host immune response and that their biosynthesis is rapidly regulated, as opposed to other components such as cytokines and chemokines, measuring their plasma concentrations may be useful in evaluating disease severity. In this context, the RECOVERY trial indicates that dexamethasone may be a useful means to regulate the uncontrolled inflammation observed in patients with severe disease [7]. Of note, we found that plasma levels of several PGs were upregulated in non-survivors when compared with survivors who were later discharged. As dexamethasone inhibits PGs [43], these results may explain a potential mechanism of action behind the benefit experienced by patients treated with such medication.

Although providing interesting insights regarding the inflammatory changes following COVID-19 infection, the present study presents some limitations. The main limit consists in it being an observational study where we were unable to evaluate directly the expression of LM biosynthetic enzymes and the activity of individual mediators found to be differentially regulated. Another limitation may be derived from the range of drugs that these patients are receiving that may influence on both enzyme activity and expression.

## Conclusions

In conclusion, this work underlines a potential role for LM in the early determination of both disease course in patients with COVID-19 and disease outcome in critically ill patients. The observations made herein suggest that the concomitant upregulation of SPM and pro-inflammatory eicosanoids in patients with severe disease reflects a more effective engagement of the host immune response than in critically ill patients. Furthermore, plasma SPM concentrations were lower in critical patients that did not survive when compared with those that were discharged, indicating that impaired engagement of pro-resolving pathways may contribute to the negative outcomes in these patients. These findings suggest that activation of LM biosynthesis is a host protective response and defects in the expression/activity of LM biosynthetic enzymes and receptors may contribute to disease severity.

## Supporting information

**S1 Table. Distinct plasma LM concentrations in patients with severe disease and critically ill patients.** The table displays the mean values ± sem of LM concentrations in pg/mL from plasma collected from COVID-19 patients with severe disease (n = 15) and critically ill patients (n = 23).
(DOCX)

**S2 Table. Plasma LM values from survivors and non-survivors patients.** The table displays the mean values ± sem of LM concentrations in pg/mL from plasma collected from COVID-19 patients who were later discharged (n = 18) or did not survive the disease (n = 4).
(DOCX)

## Author Contributions

**Conceptualization:** Jennifer Clarke, Gerard F. Curley, Jesmond Dalli.

**Data curation:** Francesco Palmas.

**Formal analysis:** Francesco Palmas, Romain A. Colas, Esteban A. Gomez.

**Funding acquisition:** Jesmond Dalli.

**Investigation:** Jennifer Clarke, Gerard F. Curley.

**Methodology:** Jesmond Dalli.

**Resources:** Jennifer Clarke, Aoife Keogh, Maria Boylan, Natalie McEvoy, Oliver J. McElvaney, Oisin McElvaney, Razi Alalqam, Noel G. McElvaney, Gerard F. Curley.

**Supervision:** Jesmond Dalli.

**Visualization:** Francesco Palmas, Jesmond Dalli.

**Writing – original draft:** Francesco Palmas, Jesmond Dalli.

**Writing – review & editing:** Francesco Palmas, Jennifer Clarke, Romain A. Colas, Esteban A. Gomez, Aoife Keogh, Gerard F. Curley, Jesmond Dalli.

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
