## [Decision Letter · Decision Letter 0]

8 Jun 2021

PONE-D-21-08835

Dysregulated Plasma Lipid Mediator Profiles in Critically Ill COVID-19 Patients

PLOS ONE

Dear Dr. Dalli,

Thank you for submitting your manuscript to PLOS ONE. After careful consideration, we feel that it has merit but does not fully meet PLOS ONE’s publication criteria as it currently stands. Therefore, we invite you to submit a revised version of the manuscript that addresses the points raised during the review process.

We look forward to receiving your revised manuscript.

Kind regards,

Fulvio D'Acquisto, PhD

Academic Editor

PLOS ONE

Journal Requirements:

Additional Editor Comments:

Reviewers' comments:

Reviewer's Responses to Questions

**Comments to the Author**

1. Is the manuscript technically sound, and do the data support the conclusions?

Reviewer #1: Yes

Reviewer #2: Yes

2. Has the statistical analysis been performed appropriately and rigorously? 

Reviewer #1: Yes

Reviewer #2: Yes

3. Have the authors made all data underlying the findings in their manuscript fully available?

Reviewer #1: Yes

Reviewer #2: Yes

4. Is the manuscript presented in an intelligible fashion and written in standard English?

Reviewer #1: Yes

Reviewer #2: Yes

5. Review Comments to the Author

Reviewer #1: This is a well-written and well-organized submission that describes a role for lipid mediators, particularly special proresolving mediators (SPMs) in the pathogenesis of SARS CoV2 infection, and the potential of using these mediators for prediction of disease outcome. The data are clear and well-presented and the conclusions of the study well-supported. The only aspect that could be improved is the description of the trial design itself in the methods. It would make the paper much clearer if the experimental design was described in some detail.

Reviewer #2: This is an outstanding scientific research report from an internationally recognized group of investigators. They report on the levels of specialized pro-resolving and pro-inflammatory lipid mediators in two groups of subjects with Covid-19: critically ill (admitted to ICU) and severe disease (hospitalized but not in ICU) using state-of-the-art liquid chromatography mass spectrometry. They found that plasma concentrations of both pro-inflammatory and pro-resolving lipid mediators were lower in critically ill patients compared to those with severe disease. Levels were lowest in those who died. Their findings establish a link between plasma lipid mediators, disease severity and survival in Covid-19.

The findings have important implications for etiology of severe versus critically ill patients with Covid-19 and treatment.

I have only a few minor comments:

line 260: "liked" should be "linked"

line 284: "the" should be "they"

line 285: Sentence starting with "while" is not a complete sentence

line 307: macrophage should be pleural

line 308: deleted "note"

Table 2: I believe this is for critically ill rather than severe patients. Severe is noted in the header.

line 566: 'ration' should be "ratio"

Table S1: It appears that the header "moderate" should be "severe" and the heading severe should be critically ill.

6. PLOS authors have the option to publish the peer review history of their article (what does this mean?). If published, this will include your full peer review and any attached files.

Reviewer #1: No

Reviewer #2: No

---

## [Author Response · Author response to Decision Letter 0]

27 Jul 2021

Reviewer #1: This is a well-written and well-organized submission that describes a role for lipid mediators, particularly special proresolving mediators (SPMs) in the pathogenesis of SARS CoV2 infection, and the potential of using these mediators for prediction of disease outcome. The data are clear and well-presented and the conclusions of the study well-supported. The only aspect that could be improved is the description of the trial design itself in the methods. It would make the paper much clearer if the experimental design was described in some detail.

We would like to thank the reviewer for their positive comments. We agree with your comments about the study design and a more comprehensive description of the study methods has now been added. We have also amended a numerical error in the WHO scores, which was noticed upon review.

Reviewer #2: This is an outstanding scientific research report from an internationally recognized group of investigators. They report on the levels of specialized pro-resolving and pro-inflammatory lipid mediators in two groups of subjects with Covid-19: critically ill (admitted to ICU) and severe disease (hospitalized but not in ICU) using state-of-the-art liquid chromatography mass spectrometry. They found that plasma concentrations of both pro-inflammatory and pro-resolving lipid mediators were lower in critically ill patients compared to those with severe disease. Levels were lowest in those who died. Their findings establish a link between plasma lipid mediators, disease severity and survival in Covid-19. The findings have important implications for etiology of severe versus critically ill patients with Covid-19 and treatment. I have only a few minor comments:

line 260: "liked" should be "linked"

line 284: "the" should be "they"

line 285: Sentence starting with "while" is not a complete sentence

line 307: macrophage should be pleural

line 308: deleted "note"

Table 2: I believe this is for critically ill rather than severe patients. Severe is noted in the header.

line 566: 'ration' should be "ratio"

Table S1: It appears that the header "moderate" should be "severe" and the heading severe should be critically ill.

We would like to thank the reviewer for their kind comments and the recognition of significance of our hard. Moreover, we would like to express our appreciation for the detailed feedback provided, which allowed us to polish the delivery of this manuscript.

We corrected the typos highlighted in your comment.

---

## [Editor Report · Decision Letter 1]

3 Aug 2021

Dysregulated Plasma Lipid Mediator Profiles in Critically Ill COVID-19 Patients

PONE-D-21-08835R1

Dear Dr. Dalli,

We’re pleased to inform you that your manuscript has been judged scientifically suitable for publication and will be formally accepted for publication once it meets all outstanding technical requirements.

Kind regards,

Fulvio D'Acquisto, PhD

Academic Editor

PLOS ONE
---

## [Editor Report · Acceptance letter]

19 Aug 2021

PONE-D-21-08835R1 

Dysregulated Plasma Lipid Mediator Profiles in Critically Ill COVID-19 Patients 

Dear Dr. Dalli:

I'm pleased to inform you that your manuscript has been deemed suitable for publication in PLOS ONE. Congratulations! Your manuscript is now with our production department. 

Kind regards, 

on behalf of

Professor Fulvio D'Acquisto 

Academic Editor

PLOS ONE